# Artificial Neural Network Based Apple Yield Prediction Using Morphological Characters

**Bharti** [1], **Pankaj Das** [1,*], **Rahul Banerjee** [1], **Tauqueer Ahmad** [1], **Sarita Devi** [2] **and Geeta Verma** [2]

1    Division of Sample Surveys, ICAR-Indian Agricultural Statistics Research Institute, New Delhi 110012, India
2    Department of Basic Sciences, Dr. YSP University of Horticulture and Forestry, Nauni-Solan 173230, India
*    Correspondence: pankaj.iasri@gmail.com

**Abstract:** The yield of the crop is a complex function of a number of dependent traits, which makes yield prediction a statistically difficult task. A number of work on yield prediction using morphological characters already exists in the literature. Most of the work used statistical techniques such as linear regression and crop yield models, which assume a linear relationship between yield and the morphological traits; in actual practice, such a linear relationship is seldom achieved. With the advancement in the field of machine learning techniques, these methods can provide a viable alternative for dealing with nonlinear relationships for yield prediction. Globally, apples are the most consumed fruit. In this paper, attempts have been made to predict the yield of the apple crop using morphological traits. PCA was used for selection of the significant variables. These variables were later used as input variables in the ANN model with different hidden layers for predicting crop yield. The predictive performance of the model was evaluated using standard statistical tests. Sensitivity analysis was performed to find out the individual effects of each character on the apple yield. The study contributes to a better understanding of the complex relationships between crop yield and morphological traits.

**Keywords:** apple fruit crop; morphological characters; yield prediction; machine learning; principal component analysis (PCA); artificial neural network (ANN)

## 1. Introduction

Apple (*Malus domestica*) is commercially cultivated in the Himachal Pradesh, Jammu and Kashmir, and Uttarakhand. These states collectively produce 99.43% of the total apple production (2,734,000 tones) in India [1]. Apple yield is a complex variable that depends on a number of factors, either directly or indirectly, including vegetative, flowering, and fruiting characteristics. Identifying a single variable representative of yield may not be reliable, so researchers are faced with the possibility of separately examining many related variables [2]. The attempt to perform a series of univariate statistical analysis for each of the variables does not hold much promise, because it overlooks the correlation among the variables and occasionally the conclusions may be deceptive. Instead, statistical methods that consider the interdependence and relative importance of several affecting factors may yield information that is more informative. Therefore, morphological characters can be considered together for yield prediction using machine-learning techniques [3–5].

For proper crop management and plan strategies for the efficient marketing of fruits, yield prediction is very crucial. The earlier the prediction, the more effectively it can be applied to improve marketing strategies. A wide range of techniques, i.e., statistical tools, crop model, and algorithms, have been developed and applied for the prediction of yield in agriculture. Correlation and multiple regression analysis are the most frequently used techniques for the prediction of yield and for the identification of important variables that affect crop yield [6–9]. However, the results are not particularly encouraging, because polynomial and interaction terms, which were not taken into account, exist [10]. Moreover,

the assumption of linear relationships between crop yield and explanatory variables is rarely met in reality, and when these relationships are not linear, the results may be deceptive [10–12]. Furthermore, principal component and factor analyses [7,13,14] can be used to lessen the issue brought on by interdependent variables, make it easier to understand complex relationships, and decrease the dimensionality of the dataset by selecting the most appropriate subset of variables that significantly affect the response variable [7]. For capturing the nonlinearity and complicated interaction between the variables, machine learning approaches such as artificial neural networks (ANNs) are emerging as an alternative to conventional linear models. ANN is a nonlinear data-driven method that follows a self-adaptive learning approach [15]. By analyzing a large number of input and output instances, ANNs discover relationships to create a formula that can be used for predictions. The development of models using ANN does not call for any prior knowledge of the inputs and outcomes. ANN is also better than any other linear model because it is also more capable of determining the optimal pattern of variables and offers less inaccuracy [16]. As a result of these benefits, ANN is very well-liked in a variety of fields, including hydrology and agriculture [3,12,13,17–20].

The majority of modelling research on crop yield prediction assumes a linear relationship between yield and its contributing characters, and is thus centered on linear regression, step-wise regression, path analysis, principle component (PCA), and factor analyses, etc. These techniques would decrease the number of variables, but they would not be sufficient or thorough enough to capture the highly nonlinear and complicated relationships between yield and other characteristics. Consequently, artificial neural networks (ANN) would be appropriate when the variables under consideration have complex and nonlinear relationships. Correlated input causes confusion for the neural network during learning, which is the key factor affecting a neural network's performance. In addition, an ANN model with a large number of input variables may perform poorly in terms of generalization [21,22]. The principal component analysis and ANN can be used together to address these problems. The present study was conducted with the aim to build up and evaluate the predictive performance of PCA-based ANN models to predict the apple yield using morphological characters as the input variables. PCA was employed for the selection of variables (feature selection), which were used as input variables in ANN models for the prediction of yield. The results from the study will help to identify and model the complex relationship between apple yield and its related morphological characters.

## 2. Materials and Methods

### 2.1. Study Area and Data Description

The study was conducted in a commercial farmer's apple orchard located at an elevation of 1901 m in Jubbal, Shimla, Himachal Pradesh (31°10′ N, 77°66′ E) during 2014–2015. The region is generally cool throughout the year with temperatures ranging from 15–25 °C during the summer and falling below zero degrees during the winter. A representative sample of trees were selected from the experimental orchard, and four branches from each of the tree in four directions as per the practice in vogue were selected for recording the observations on various morphological characters, i.e., plant height, canopy spread, plant girth, flower density (FD), flower density index (FDI), flowering intensity (FI), fruit set (FS), crop density (CD), and length diameter ratio (LD ratio) of variety Royal delicious were recorded. Data on vegetative characteristics were collected during April month. Data on flowering and fruiting characteristics were collected during months of May, July, and August. The yield of apple was recorded by collecting the fruit manually from each tree. Summary statistics of each characters are presented in Table 1.

**Table 1.** Summary of plant parameters.

| Characters | Range | Mean | Std. Deviation |
|---|---|---|---|
| Plant height (m) | 3.05–11.89 | 7.22 | 2.21 |
| Canopy spread (m) | 1.32–9.48 | 5.57 | 2.03 |
| Plant girth (cm) | 0.15–0.91 | 0.61 | 0.18 |
| Flower density | 1.00–10.82 | 3.54 | 1.99 |
| Flower density index | 0.10–1.08 | 0.35 | 0.18 |
| Flowering intensity | 0.35–0.50 | 0.41 | 0.03 |
| Fruit set | 0.15–0.57 | 0.31 | 0.08 |
| Crop density | 0.30–4.40 | 1.09 | 0.66 |
| Length diameter ratio | 6.84–10.28 | 8.22 | 0.68 |

The selection of appropriate input variables for the development of MLR and ANN models is very crucial [23,24]. Although much research has employed the simple correlation as an input selection approach [17,25], this method is unable to reveal the types of direct or indirect effects between variables. Furthermore, it also reduces the probability of having a unique solution [26]. Principal component analysis is a data reduction, which can be used to select the most important uncorrelated variables as the input variables.

### 2.2. Development of Artificial Neural Network Model

An artificial neural network (ANN) is a type of machine learning model that is a data-driven nonlinear adaptive learning method [15]. An ANN model network can capture the representation of complex data patterns, which are difficult to model either with traditional model-based approaches or knowledge-based expert systems [27]. A typical ANN model consists of three main layers, i.e., input layer, hidden layer, and output layer (Figure 1). These layers contain simple processing units that are known neurons or nodes. The nodes are interconnected to each other through weighted connection, which varies according to the specified architectures of required ANN model [27]. The number of hidden layers and its nodes depend on the specific problems of the study. Several studies [16,28,29] have suggested that the trial and error method is the most common method to find an optimum number of hidden layers and its nodes. Further details of the ANN model and its application are given in Haykin (2008) [30]. The output of ANN model can be expressed by following equation (Equation (1)) [18]:

$$y_t = \alpha_0 + \sum_{j=1}^{n} \alpha_j f(\sum_{i=1}^{m} \beta_{ij} y_{t-1} + \beta_{oj}) + \varepsilon_t \tag{1}$$

where $y_t$ is the output of the neural network model (yield per plant), $n$ is number of hidden nodes, $m$ is the number of input nodes, f is the net input of the activation function, $\beta_{ij}$ $\{i = 1, 2, \ldots, m; j = 0, 1, \ldots, n\}$ are the weights from input to hidden nodes, $\alpha_j \{j = 0, 1, \ldots, n\}$ are the vectors of the weights from the hidden to output nodes, and $\alpha_0$ and $\beta_{0j}$ are the weights of arcs leading from bias terms. Activation function is a differentiable function that is used for smoothing the result of the cross product of the covariates or neurons and the weights. In the artificial neural networks, the activation function of a node defines the output of that node given an input or set of inputs.

In the present study, multi-layered feed-forward network architecture with a logistic function was used as an activation function. The Levenberg–Marquardt (LM) learning algorithm was used to adjust the weights in the multi-layered feed-forward networks. To obtain the best topology of the ANN model, different numbers of hidden layers (1–6) and nodes in each hidden layer (1–20) were tested using the trial and error method. The epoch (iterations) size and mean square error (MSE) threshold values for each run in the training and cross validation dataset were 100 and 0.01, respectively. The convergence of the average MSE values during training and cross validation was investigated at an epoch length of 0 to 100 to avoid model over-fitting and memorization. It was investigated that

the convergence point was epoch 60 so as to avoid over-fitting (Figure 2). The dataset was portioned into two subsets, i.e., training set (80%) and testing set (20%) for fitting of ANN and MLR model.

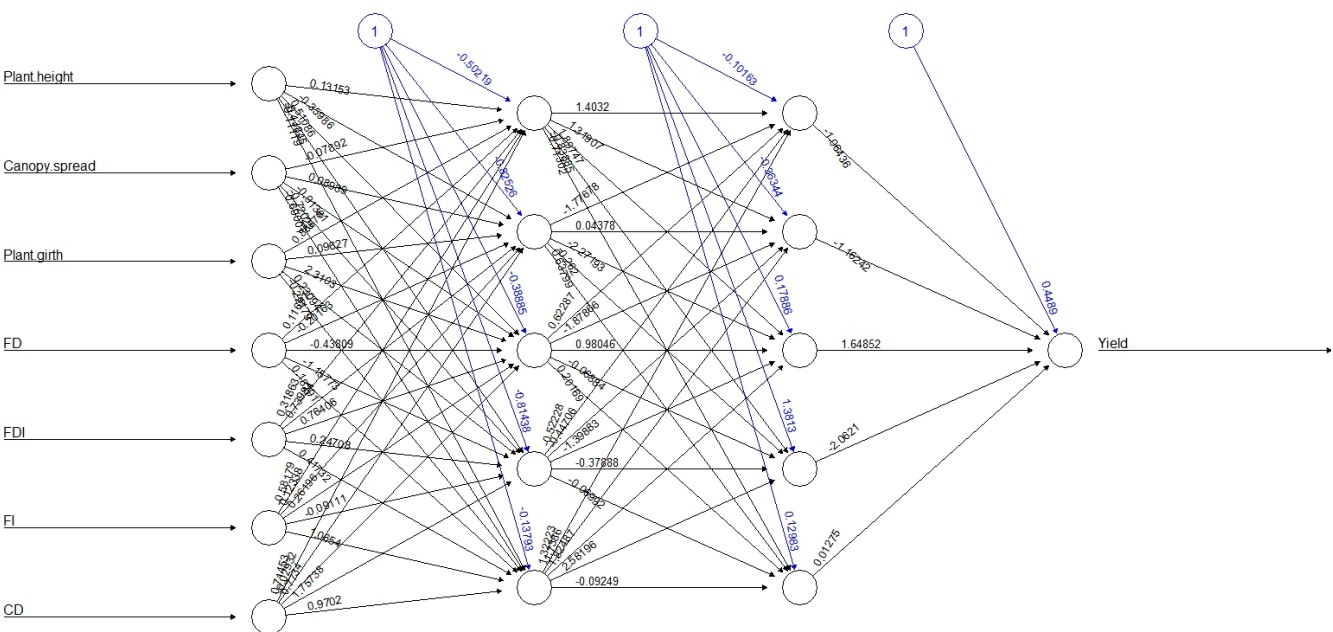

**Figure 1.** Topology of neural network model for Apple yield prediction.

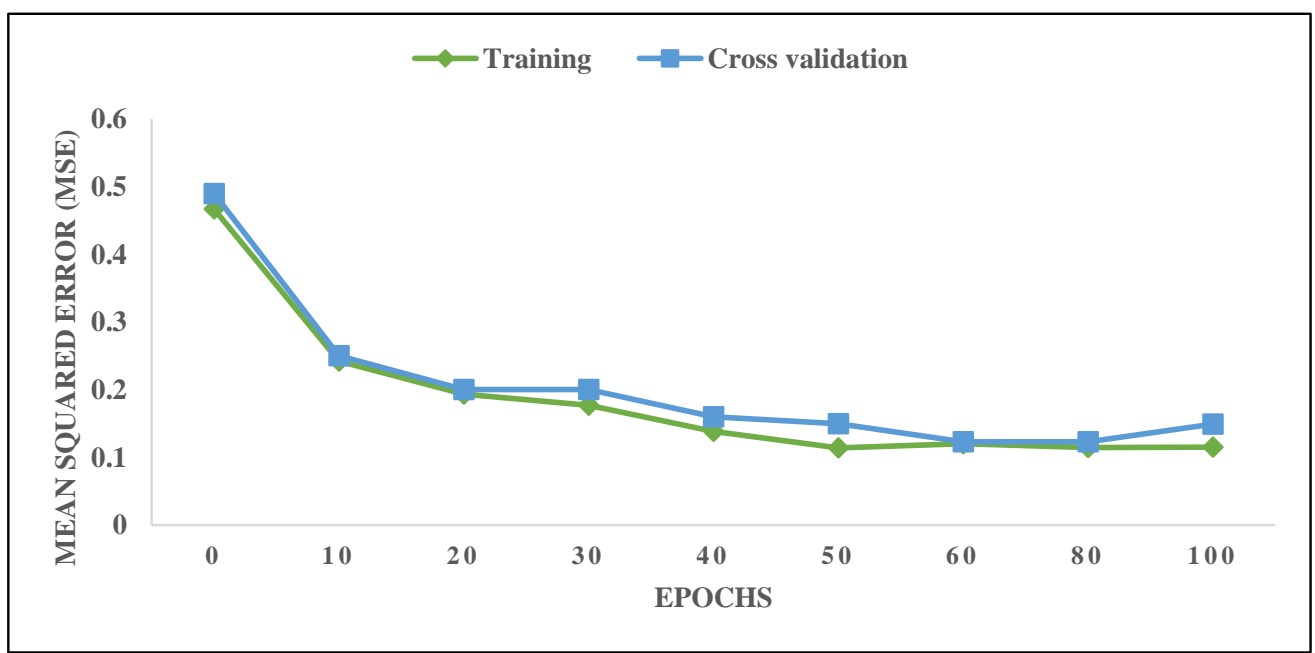

**Figure 2.** Convergence of the average MSE value during training and cross validation of ANN model.

*2.3. Development of Multiple Linear Regression Model*

Crop yield prediction in agriculture has made extensive use of regression-based models. Multiple linear regression (MLR) attempts to model the relationship between the regressed and more than one regressor [31,32]. In MLR, an attempt is made to account for

the variation of the regressors in the regressed synchronically [33]. The model for MLR can be written as follows:

$$y = \beta_0 + \beta_1 x_1 + \beta_2 x_2 + \ldots + \beta_k x_k + \varepsilon \tag{2}$$

where $y$ denotes regressed (yield), $x_i$ denotes regressors (morphological characters), and $\varepsilon$ is the error term which is normally distributed with zero mean and constant variance.

### 2.4. Model Performance Measures

The performance of the fitted models were evaluated using four statistical measures, including root mean square error (RMSE), mean absolute deviation (MAD), mean absolute percentage error (MAPE), and coefficient of determination ($R^2$) [34]. The functional formula of these measures were used as follows

$$RMSE = \sqrt{\frac{\sum\limits_{i=1}^{N}(y_i - \hat{y}_i)^2}{N}} \qquad MAD = \frac{\sum\limits_{i=1}^{N}|y_i - \hat{y}_i|}{N}$$

$$MAPE = \frac{\sum\limits_{i=1}^{N}|y_i - \hat{y}_i|/y_i}{N} \qquad R^2 = \frac{\sum\limits_{i=1}^{N}(y_i - \overline{y})(\hat{y}_i - \overline{\hat{y}})}{\sqrt{\sum\limits_{i=1}^{N}(y_i - \overline{y})^2 \sum\limits_{i=1}^{N}(\hat{y}_i - \overline{\hat{y}})^2}} \tag{3}$$

where $y_i$ *and* $\hat{y}_i$ and are the actual value and predicted value of response variable and $N$ is the number of data.

## 3. Results

### 3.1. Selection of Input Variables

The input variables form the model structure, have an impact on the weighted coefficient, and influence the results of the models; this is why their selection is a key factor in any modelling method [24,35]. A simple correlation coefficient can be used as the input variable selection method as it can identify the characters, which have a strong correlation with the output variables (Figure 3).

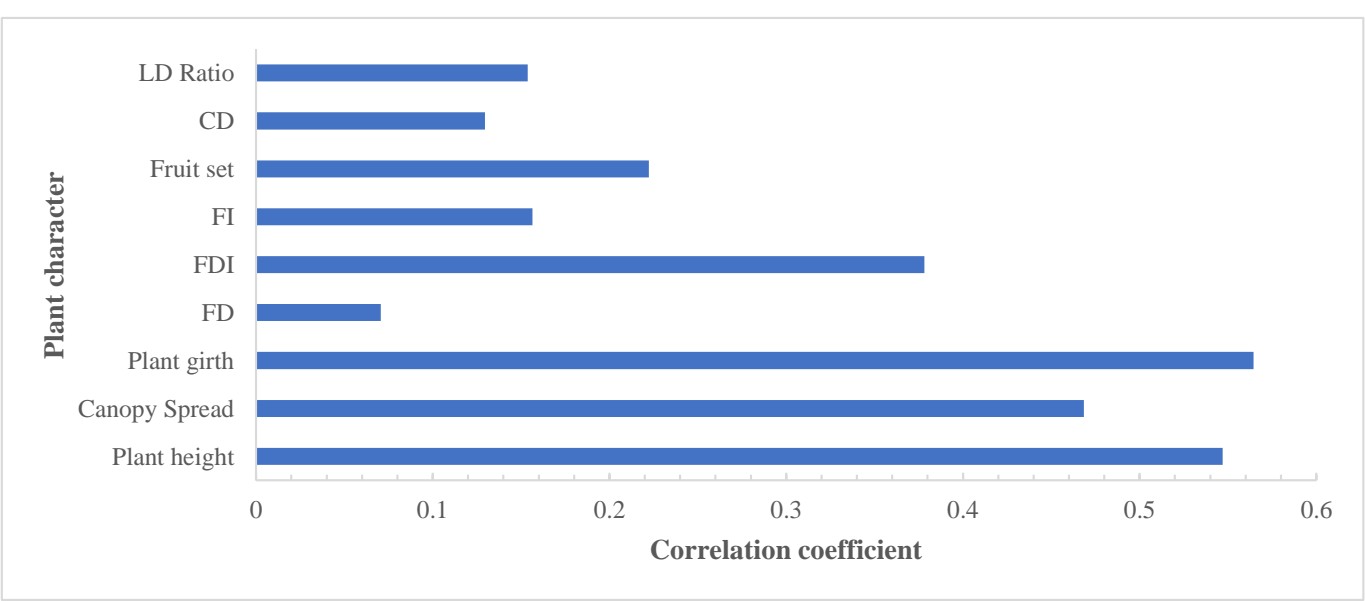

**Figure 3.** Correlation plot of different input variables.

There is a positive and significant correlation between apple yield and plant height, canopy spread, plant girth, FDI, and fruit set. Similar findings were reported by other researchers [19,36–38]. Plant height, canopy spread, plant girth, FDI, and fruit set were

regarded as the most significant characters based on the correlation analysis, as they have a significant positive association with yield.

Principal component analysis (PCA) can be used as a more efficient method of input variable selection than correlation analysis, as a simple correlation coefficient between characters can be influenced by the positive or negative indirect effect of another variables [24,26]. The PCA results (Table 2) helped to find the explained variations using the morphological characters. Together, first and second principal components explained 56.53% of the total variation in the variables. According to these principal components, plant height, canopy spread, plant girth, FD, FDI, FI, and CD were identified as most appropriate input variables. Consequently, a suitable combination of these variables accompanied by high-performance modelling can result in an effective model to predict yield.

**Table 2.** Principal component analysis for input variable selection.

| Characters | PH | CS | PG | FD | FDI | FI | FS | CD | LDR | EV | CV |
|---|---|---|---|---|---|---|---|---|---|---|---|
| PC1 | −0.4223 | −0.4222 | −0.4178 | 0.3909 | 0.2924 | 0.1073 | 0.1183 | 0.4389 | 0.1108 | 3.07 | 34.15 |
| PC2 | 0.3905 | 0.3051 | 0.3672 | 0.3810 | 0.4114 | 0.4333 | −0.0969 | 0.3284 | −0.011 | 2.01 | 56.53 |

PH: plant height, CS: canopy spread, PG: plant girth, FD: flowering density, FDI: flowering density index, FI: flowering intensity, FS: fruit set, CD: crop density, LDR: LD ratio, EV: eigen value, CV: cumulative variance.

### 3.2. ANN Model Development

In the present study, the analysis of the dataset was carried out in RStudio. The multi-layered feed-forward network architecture with different functions was used for yield prediction. Seven plant characters *viz.* plant height, canopy spread, plant girth, FD, FDI, FI, and CD were found significant using PCA. These plant characters were used as input variables based on the variable selection method for ANN model fitting. Based on the significant advantage of Levenberg–Marquardt compared with other optimization algorithms, this algorithm was used in all ANN models. The performance measures, such as RMSE, MAD, MAPE, and $R^2$, were considered for the evaluation of the model's performance. The performance of different activation functions was also tested. It has been observed that logistic activation outperformed among the others due to its ability to capture nonlinear variation in the dataset. Ahmadi et al. [39], Hagan et al. [40], and Mansouri et al. [17] also reported the ability of nonlinear functions to cover nonlinear patterns in a dataset. The different number of hidden layers with a different number of nodes were fitted to obtain the best topology for the neural network model (Table 3). The results indicated that the ANN model with two hidden layers (5-5), i.e., 7-5-5-1 architecture provide best result. This ANN model (7-5-5-1) had the lowest RMSE, MAD, and MAPE values with the highest model accuracy in both the training and testing stages. The schematic diagram of the ANN structure (7-5-5-1) is presented in Figure 1.

**Table 3.** The performance of ANN models with different hidden layers in the training and testing set.

| | Hidden Layer | Best Topology | RMSE | MAD | MAPE | $R^2$ | Accuracy (%) | Error Rate |
|---|---|---|---|---|---|---|---|---|
| Training | 1 | 7-3-1 | 36.3360 | 25.7337 | 0.2306 | 0.8121 | 90.36 | 0.2422 |
| | 2 | 7-5-5-1 | 24.8300 | 18.2607 | 0.1523 | 0.9430 | 98.72 | 0.0736 |
| | 3 | 7-3-3-3-1 | 31.0590 | 22.3937 | 0.2053 | 0.8629 | 93.59 | 0.1769 |
| | 4 | 7-3-3-3-3-1 | 27.4964 | 21.2744 | 0.2136 | 0.8924 | 92.23 | 0.1386 |
| | 5 | 7-5-5-1-5-5-1 | 24.9840 | 19.8195 | 0.1556 | 0.9116 | 93.10 | 0.1140 |
| | 6 | 7-3-3-3-3-5-5-1 | 34.8300 | 17.81 | 0.2426 | 0.9113 | 95.10 | 0.11438 |
| Testing | 1 | 7-3-1 | 63.2026 | 43.9649 | 0.3582 | 0.5622 | 93.01 | 0.2422 |
| | 2 | 7-5-5-1 | 36.6078 | 28.1045 | 0.2151 | 0.8685 | 95.36 | 0.0736 |
| | 3 | 7-3-3-3-1 | 52.2906 | 38.2418 | 0.2974 | 0.7129 | 91.32 | 0.1769 |
| | 4 | 7-3-3-3-3-1 | 43.7711 | 28.2788 | 0.1900 | 0.7935 | 93.49 | 0.1386 |
| | 5 | 7-5-5-1-5-5-1 | 40.0703 | 28.2700 | 0.2111 | 0.8239 | 92.65 | 0.1140 |
| | 6 | 7-3-3-3-3-5-5-1 | 43.2684 | 32.92371 | 0.2360 | 0.8073 | 89.33 | 0.1144 |

The topology was able to express approximately 94% variability in the training phase and approximately 86% in the testing phase. The scatter plot of the measured and predicted yield of the apple in testing is represented in Figure 4a. The results show that the distribution of the predicted apple yield had a close distribution with the actual apple yield. Further boxplot (Figure 4b) results showed there was no outlier in the predicted data, which was an indication of proper model fitting. Balas et al. [41] suggested PCA, as variable selection (data pre-processing) reduced the chance of over-fitting. The application of ANN with PCA (PCA-ANN) increased the forecasting ability of the ANN model compared with a single ANN model [16].

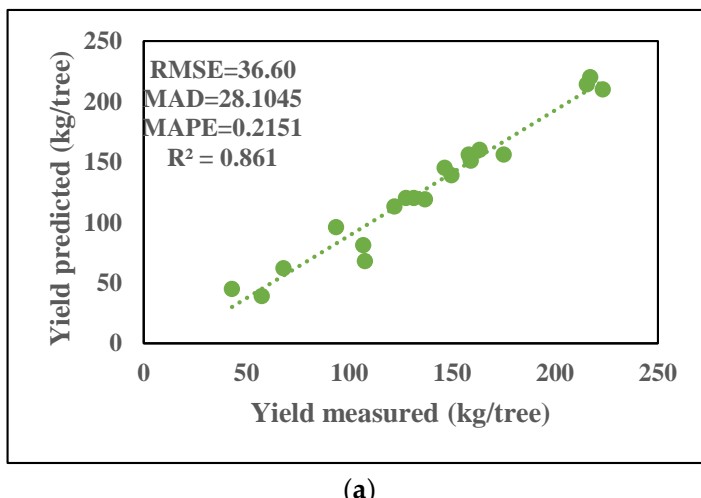
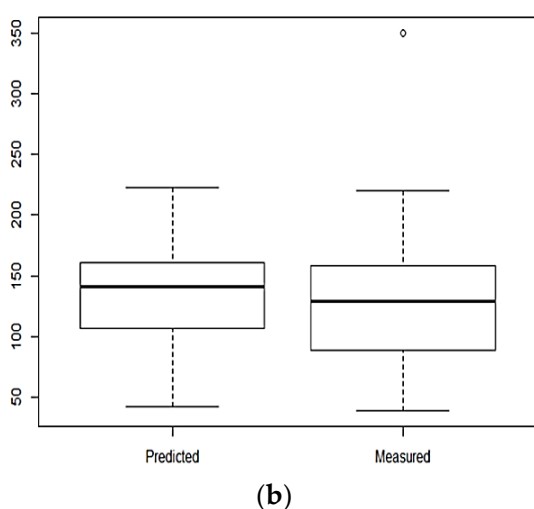

(**a**)                                                                      (**b**)

**Figure 4.** (**a**) Scatter plot of the measured and predicted yield of apple in the testing stage of ANN; (**b**) boxplot of measured and predicted apple yield in the testing stage of ANN. Green dots denote the observations and root mean square error (RMSE), mean absolute deviation (MAD), mean absolute percentage error (MAPE), and coefficient of determination ($R^2$) are the performance measures.

### 3.3. MLR Model Development

Multiple linear regression (MLR) model is a commonly used method for crop yield prediction. In the present study, the same data variables that were used for the ANN model were used for MLR model building. The fitted regression model to predict the apple yield was as:

$$Yield = -0.347 + 0.186 * Plant\ height + 0.27 * canopy + 0.441 * FDI \qquad (4)$$

Equation (4) shows that the predicted value of the apple yield is a linear combination of other significant variables (plant height, canopy spread, and FDI). It also helped to see how the prediction value of the apple yield changed with the unit change in the variables (plant height, canopy spread, and FDI). The results also showed that the MLR model had a low $R^2$ value (70.69%) in Figure 5a. The scatter plot indicated that the MLR model did not cover all of the data points and most of the data points deviated from the regression line. Further boxplots (Figure 5b) of the measured and predicted apple yield in the testing stage of MLR indicate the inefficiency of the MLR model to predict apple yield.

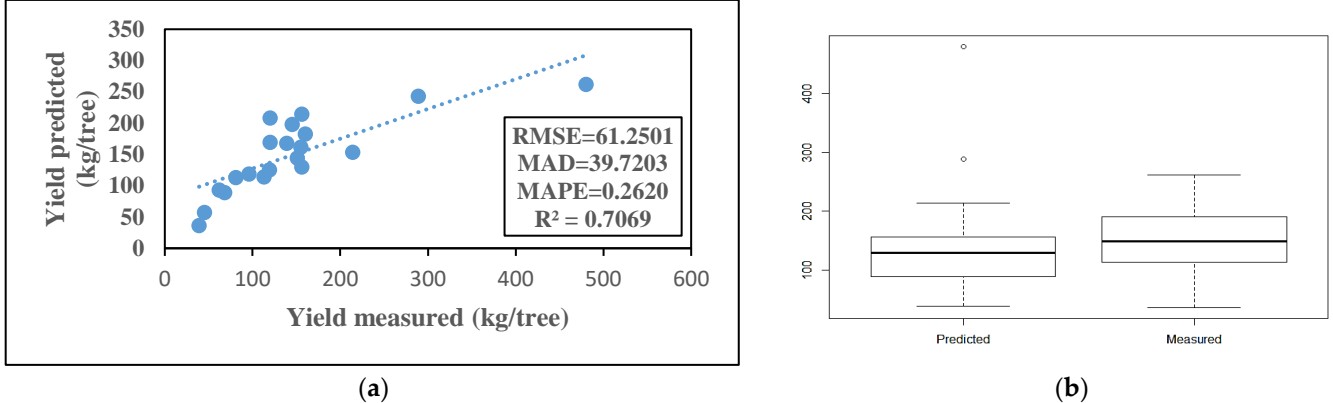

(**a**)                                               (**b**)

**Figure 5.** (**a**) Measured and predicted apple yield in testing stage of MLR; (**b**) boxplot of the measured and predicted apple yield in the testing stage of MLR. Blue dots denote the observations.

## 4. Discussion

### 4.1. Comparison of Fitted Models

The prediction performance of the MLR and ANN models was evaluated for statistical measures such as RMSE, MAD, MAPE, and $R^2$. The results are presented in Table 4. It has been observed that the selected ANN model outperformed with an 18.60% increase in $R^2$ and a reduction of 67.31%, 41.33%, and 21.80% in RMSE, MAD, and MAPE compared with the MLR model.

Besides these measures, a graphical representation (Figure 6) of the actual and predicted by the ANN and MLR model helped to understand the superiority of the ANN model over the MLR. The ANN model captured the data pattern more accurately than the MLR model. The possible reason behind the poor performance of the MLR model is the nonlinear portion of the relationship between the input variables and apple yield. The MLR model did not capture this relationship, while the ANN model took this relationship into account during model building. These results indicate how selection of the proper model improves the prediction of the dependent variable (yield). ANN models have a high ability to model nonlinear and complex relationships among the data variables compared with MLR models. In the literature, similar results have been reported in many studies [16,20,42].

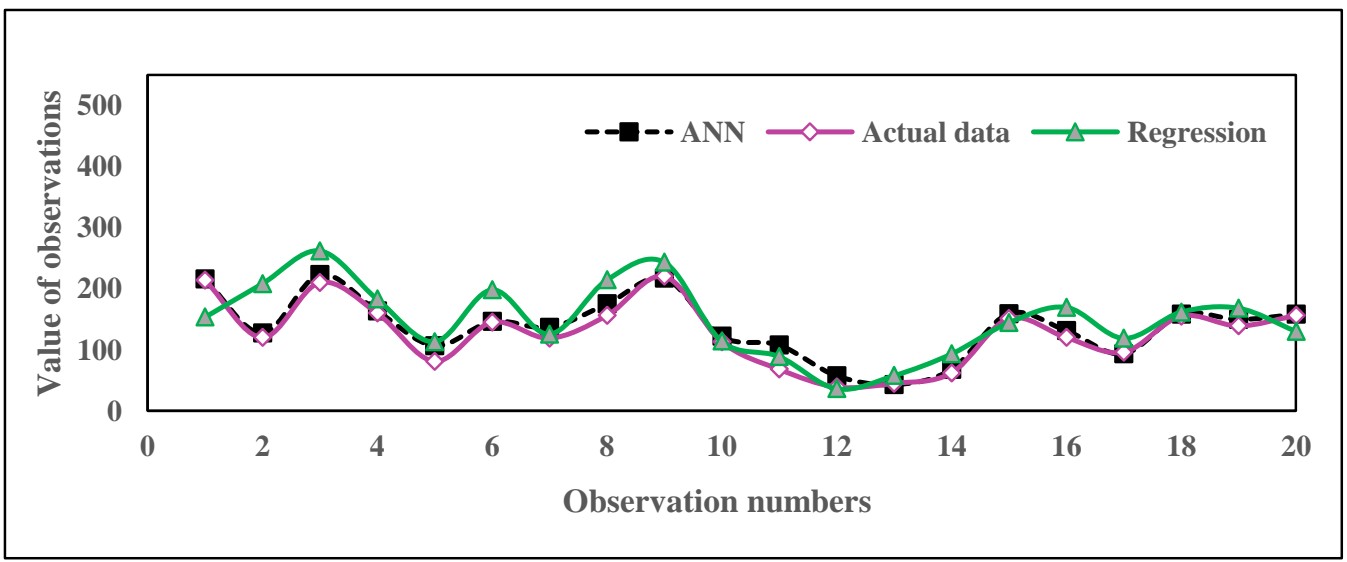

**Figure 6.** Predicted values of the fitted models with actual data points.

**Table 4.** Prediction performance of the fitted models in the testing stage.

| Model | RMSE | MAD | MAPE | $R^2$ |
|---|---|---|---|---|
| ANN | 36.6078 | 28.1045 | 0.2151 | 0.8685 |
| MLR | 61.2501 | 39.7203 | 0.2620 | 0.7069 |

*4.2. Sensitivity Analysis*

A sensitivity analysis was performed to find out the individual effects of each of the input variables on the prediction values of apple yield. The results (Figure 7) indicate how the performance of the ANN models change with different combinations of input variables (plant height, canopy spread, plant girth, FD, FDI, FI, and CD) [2].

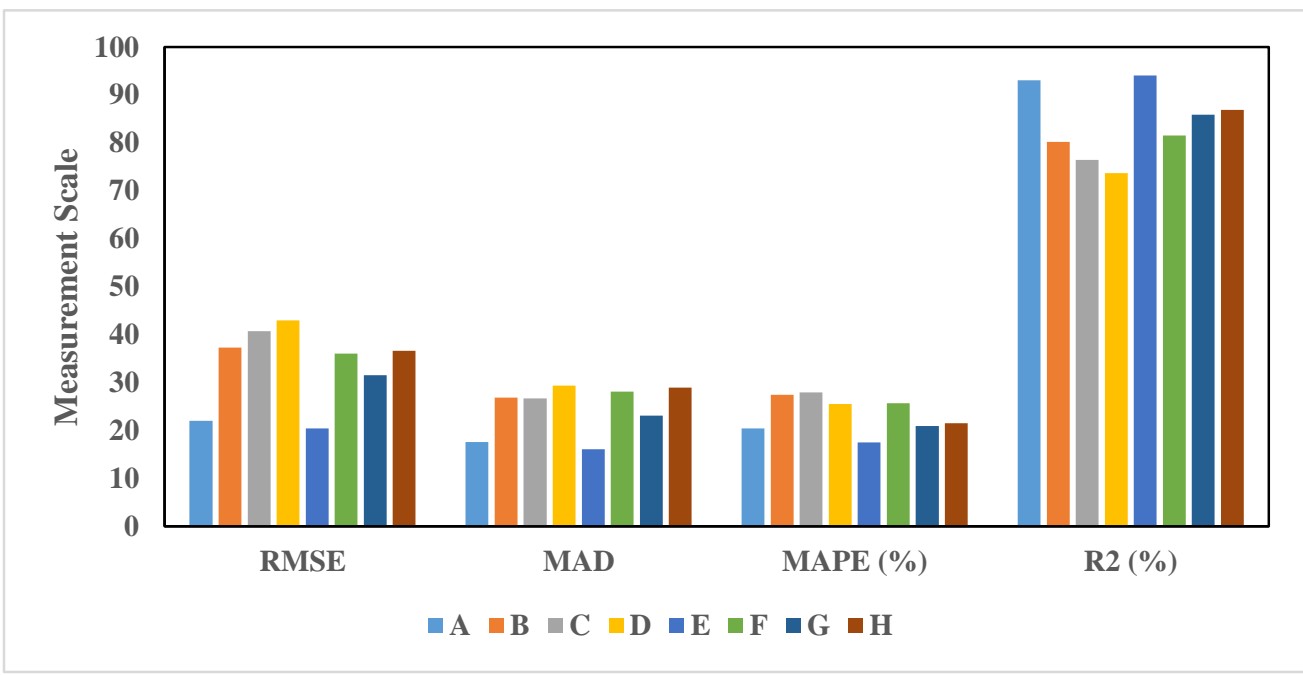

**Figure 7.** Sensitivity analysis of input variables on apple yield in ANN model. A: The best ANN model without CD; B: The best ANN model without FI; C: The best ANN model without FDI; D: The best ANN model without FD; E: The best ANN model without plant girth; F: The best ANN model without canopy spread; G: The best ANN model without plant height; H: The best ANN model (with plant height, canopy spread, plant girth, FD, FDI, FI, and CD as the input).

The ability of ANN to predict apple yield was significantly decreased when it was run without FI, FDI, and FD. The models without FD have the lowest $R^2$ and highest RMSE (79.47), MAD (79.44), and MAPE (23.07). Therefore, FD can be considered as an influential factor to predict apple yield. In addition to these characteristics, FDI and FI also had a significant effect on predicting apple yield.

**5. Conclusions**

Yield prediction of fruit crop based on the morphological characters, is a beneficial approach. In the present study, primary data on apple crop yield, as well as the morphological characters viz. plant height, canopy spread, plant girth, flower density, flower density index, flowering intensity, fruit set, crop density, and length diameter ratio was collected from commercial apple growing farmers in Jubbal block, Himachal Pradesh, India. A simple correlation was run to see the impact of the morphological characters on the yield, and it was observed that there was a positive and significant correlation between apple yield and plant height, canopy spread, plant girth, FDI, and fruit set. A PCA was run to select the significant variables, as the selection of variables is crucial for model building.

First and second principal components explained 56.53% of the total variation in the variables, plant height, canopy spread, plant girth, FD, FDI, FI, and CD were identified as the most appropriate input variables. Hence, a combination of these variables was used as the input variables for model building. A multi-layered feed-forward network architecture with different functions was used for model building and yield prediction. Seven plant characters, i.e., plant height, canopy, tree girth, FD, FDI, FI, and CD were used as input variables based on the variable selection method for ANN model fitting. The Levenberg–Marquardt algorithm was used in all of the ANN models. The model performance was evaluated using standard statistical measures such as RMSE, MAD, MAPE, and $R^2$. The logistic activation function was found to outperform all other activation functions. This ANN model (7-5-5-1) had the lowest RMSE, MAD, and MAPE values with the highest model accuracy in both the training and testing stages. Furthermore, the results show a close association between the predicted and actual yield of apple. As MLR models are predominantly used in crop yield prediction, the MLR model was also used for the study and it was observed that the selected ANN model outperformed the MLR model with an 18.60% increase in $R^2$ and a reduction of 67.31%, 41.33%, and 21.80% in RMSE, MAD, and MAPE. All of the computations have been carried out by writing suitable codes in R software available with the authors.

**Author Contributions:** B. and P.D. conceived the conceptualization, investigation, formal analysis, data curation, and writing original draft. G.V. and S.D. gave the idea of resources, reviewing, data collection, and editing. R.B. and T.A. performed supervision and edited the manuscript. All authors have read and agreed to the published version of the manuscript.

**Funding:** This research was funded by the ICAR, Indian Agricultural Statistics Research Institute, New Delhi, India.

**Data Availability Statement:** The datasets analyzed during the current study are available from the corresponding author upon reasonable request.

**Acknowledgments:** The authors are thankful to ICAR-IASRI for providing facilities for carrying out the present research and to the farmers for their co-operation in data collection.

**Conflicts of Interest:** The authors declare no conflict of interest.

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
