# Peer review of "Artificial Neural Network Based Apple Yield Prediction Using Morphological Characters"

_horticulturae, doi:10.3390/horticulturae9040436_

Round 1

Reviewer 1 Report

The manuscript uses the artificial neural network models (ANNs) as a system for predicting crop yields. This type of studies based on ANNs have been applied in many fields, such as river flow prediction, modeling and control of nonlinear systems, prediction of critical micellar concentration values in different surfactants, or even in stock market prediction systems, …, developing predictive models based on variety, type of production and harvesting dates (Hernandez et al., 2015 DOI:10.1371/journal.pone.0128566; Huang et al., 2021DOI: 10.1002/fsn3.2166). The authors apply the ANNs methodology to create a predictive yield model based on morphological characters.

INTRODUCTION

Los autores deberían ampliar en este apartado la importancia y beneficios de la utilización de redes neuronales y su utilización para el diseño de modelos predictivos basados en una gran cantidad de variables.

¿Cómo se han incluido las referencias? La primera referencia, line 26 aparece como  [10], la segunda en line 30 es la [26]

The authors should expand in this section on the importance and benefits of the use of ANNs and their usage for the design of predictive models based on a large number of variables.

How have the references been included? The first reference, line 26, appears as [10], the second on line 30 is [26].

MATERIAL AND METHODS

In which year or years was the sampling carried out? It is not indicated whether it is a single year or several years. Is the study one year? Why? Results may not be reliable.

In the references cited, line 80 for example, it appears as [1 and 28] and it should be [1,28]. Review the entire text.

In Table 1, in the characters column, either full name or abbreviation appears indistinctly: they should rather be unified.

RESULTS

The variables used were selected by means of PCA. From Table 2, the variables used for the neural network have been selected (Figure 1, Figure 2). However, variables appear in the neural network that are not among those of the PCA (“weight”). Why? Is it Plant height?

The accumulated variance barely explains 56% of the variability of the data used. However, the ideal goal would be for PC1 to accumulate more than 65-70 %.

Line 124: models () ‘’

Line 146: the figure 2 is really the figure 3

Figure 2 (really figure 3) line 145-146: 11 correlation coefficients appear, but only 6 plant characteristics are cited, including yield. Why are they selected? Have more variables been measured? Where?

Line 149, is weight the PH or is it a different variable not previously indicated?

Figure 4a: ((kg/tree) change to (kg/tree)

DISCUSSION

Is no relationship observed between FI and FS?

REFERENCES

Review the references, the last citation is not numbered. Homogenize the abbreviation of the journals: Soc and Soc.

Author Response

Respected Sir,

Reviewer 2 Report

The work concerns the prediction of apple yield on the basis of morphological features of apple trees.
The authors compared the model of linear regression with a model based on PCA and ANN. Undoubtedly, these methods can be a viable alternative
in dealing with non-linear relationships in yield prediction.

The work is written in the correct language and is included in the subject of the journal.

There are many places where abbreviations are used that should be explained.
e.g. Table 1 it should be self explanatory

L24: Malus domestica Latin names should be italicized
L26: please add "in India"

General comment:

Did the data analyzed at work come from only one year?
Independent variables influencing the yield come from the characteristics of the apple tree.

100 trees of one variety were included in the study.
In my opinion, it is not enough to recommend a method based on neural networks in particular.

Neural networks require many cases (several thousand at best).
For a hundred cases, the network will learn the answers by heart for them.
The authors can try, for example, CART, but there are also few cases for this method.

Author Response

Respected Sir,

Round 2

Reviewer 1 Report

The MS has been revised and modified. The introduction has been implemented and the tips included.

Reviewer 2 Report

The work has been improved. I still believe that one year is not enough to draw such conclusions. The authors only considered 100 trees. The rationale is that they are representative of all apple trees in this region. Perhaps the work will have some value for the apple cultivation in the study region.